

# The HOOPLA toolbox: a HydrOlOgical Prediction LAboratory to explore ensemble rainfall-runoff modeling

Antoine Thiboult[1], Gregory Seiller[2], Carine Poncelet[3], and François Anctil[1]

[1]Dept. of Civil and Water Engineering, Université Laval, Québec, Canada
[2]Formerly at Université Laval, Québec, Canada, now at DHI, Nantes, France
[3]Formerly at Université Laval, Québec, Canada, now at Crédit Agricole Sud Rhône Alpes, Grenoble, France.

**Correspondence:** Antoine Thiboult (antoine.thiboult@gci.ulaval.ca)

**Abstract.** This technical report introduces the HydrOlOgical Prediction LAboratory (HOOPLA) developed at Université Laval for ensemble lumped hydrological modelling. HOOPLA includes functionalities to perform calibration, simulation, and forecast for multiple hydrological models and various time steps. It includes a range of hydrometeorological tools such as calibration algorithms, data assimilation techniques, potential evapotranspiration formulas and a snow accounting routine. HOOPLA

is a flexible framework coded in MATLAB that allows easy integration of user-defined hydrometeorological tools. This report also illustrates HOOPLA's functionalities using a set of 31 Canadian catchments.

## 1 Introduction

In the 70s, meteorologists realized the importance of addressing uncertainty in forecasting to achieve more accurate predictions, but it is only in the 90s that ensemble meteorological forecasting was more broadly developed to account for the chaotic nature

of the atmospheric processes (Molteni et al., 1996; Buizza et al., 1999). Following on the success, hydrological forecasters adopted the concept of ensemble and gradually integrated probabilistic meteorological forecasts into hydrological prediction systems (e.g. Pappenberger et al., 2005; Thirel et al., 2008; Cloke and Pappenberger, 2009; He et al., 2009). Yet, it appears that consideration of meteorological forcing uncertainty remains insufficient to adequately describe the hydrological uncertainty and that additional (hydrologic) sources of errors need to be taken into account (Vrugt and Robinson, 2007; Salamon and

Feyen, 2010).

In this respect, hydrological model initial conditions received increasing attention by, in particular, the means of data assimilation. By updating model states according to streamflow measurements, various data assimilation schemes proved to be able to increase forecast performance (Weerts and El Serafy, 2006; DeChant and Moradkhani, 2012; Lee et al., 2011, e.g.). Yet, Salamon and Feyen (2009) pointed out that data assimilation underestimates predictive uncertainty if the model structural error

is not explicitly accounted for, which remains a challenge in traditional data assimilation schemes.

In parallel, multimodel and model intercomparison experiments indicated that it is unlikely that a single structure may correctly describe rainfall-runoff uncertainty (Clark et al., 2008; Pechlivanidis et al., 2011) and that the combination of several models is often superior to individual ones, in addition to providing a proxy for assessing related uncertainty (Ajami et al., 2006;





Breuer et al., 2009). Additionally, Poulin et al. (2011) showed that resorting to a multimodel approach implicitly accounts for
parameter uncertainty as multimodel ensemble spread encompasses multi-parameter ensembles.

While the necessity to address all sources of uncertainty is now recognized (Pappenberger and Beven, 2006), the vast
majority of hydrological frameworks such as SUMMA (Clark et al., 2015a, b, c), fuse (Vitolo et al., 2016), RAVEN (Craig
and the Raven development team, 2019), or SUPERFLEX (Fenicia et al., 2011; Kavetski et al., 2011) focus on the structural
uncertainty. HOOPLA was primarily designed to disentangle the three main sources of uncertainty in the hydrometeorological
modelling chain (meteorological forcing, hydrological initial condition, and structure uncertainty). By resolutely adopting a
probabilistic approach to handle the various sources of uncertainty, HOOPLA is an attempt to produce accurate and reliable
streamflow predictions without resorting to post-processing techniques (Thiboult et al., 2016). The toolbox was subsequently
recoded for a better user experience, more flexibility, and to allow an easy integration of additional hydrometeorological tools.

The scope of this paper is to present HOOPLA's main functionalities and to illustrate its performance on a set of 31
Canadian catchments. Therefore, the paper does not aim at providing the user with the knowledge to run computations with
HOOPLA. Interested readers are referred instead to the user manual for complete guideline regarding the usage of HOOPLA
(Thiboult et al., 2018). The manual and the HOOPLA software are freely available under the BSD 2-Clause License at:
https://github.com/AntoineThiboult/HOOPLA. Section 2 presents the HOOPLA's main components, Section 3 shows typical
results, and a summary is provided in Section 4.

## 2   An overview of HOOPLA's functionalities

### 2.1   Toolbox development philosophy

HOOPLA has been designed to fulfill three major requirements: to provide a user-friendly environment, to be flexible regarding
user's modeling preferences, and to allow easy integration of tools created by the user for testing or development purposes. To
ensure an easy start, a comprehensive manual (provided as supplementary material) comes along with the toolbox. The manual
provides all the necessary information to implement HOOPLA. In particular, it provides workflows for the different uses, lists
data requirements, and describes input and output formats as well as all the options that can be accessed from the main script.
The toolbox also includes a set of demo data to get familiar and can be run as is.

- User-friendly environment: HOOPLA can be easily controlled from a master script named launch_HOOPLA.m located
  at the root of the toolbox. This is the only file that needs to be edited and executed by the user for traditional applications.
All modeling options can be accessed by specifying the different fields of the structure named "Switch". For instance,
  the user may specify to run HOOPLA in calibration, with the root-mean-square error as objective function, to choose a
  three-hour time step, to export the calibrated parameters, and to loop over all catchments and models by solely editing
  the corresponding launch_HOOPLA.m script lines. This file is thoroughly documented and can be intuitively modified.
  A snippet of the master file is provided in the Appendix A. Therefore, users need only little knowledge about hydrology
and programming for typical use.



- A large number of options: HOOPLA provides a broad array of hydrometeorological tools (hydrological models, potential evapotranspiration formulas, data assimilation schemes, calibration algorithms) that can be selected by the users to meet their needs. These functionalities are documented below in this section. These tools can be used in several modes, i.e. calibration, simulation, or forecast with a flexible time step. Additionally, several of these tools possess options for fine-tuning (management of the different types of uncertainties, criteria for convergence, etc.)

- A flexible framework: The toolbox is developed in a way that allows easy integration of additional tools, providing that users possess an intermediate MATLAB knowledge. Since HOOPLA was developed as a framework that connects many modules (hydrological tools), users need to perform only three tasks to add a new feature: 1) respect input and output conventions that are inherent to the feature type, 2) drop the function file in the appropriate directory, and 3) edit the appropriate configuration file to inform HOOPLA that a new feature is available. For example, users who study parameter optimization would save time and efforts since they can solely focus on the development of calibration algorithms and implement their codes within HOOPLA. This way, they will be able to easily test their new methods on several models and catchments in calibration, simulation (validation), and forecast without the need to develop a testing framework. Additionally, HOOPLA can be used as a benchmark as the users can compare their findings to already existing calibration methods that are already present in the toolbox. A guideline with examples is provided in the user manual that details the procedure to implement new tools to HOOPLA.

Thanks to these attributes, HOOPLA is polyvalent and may be used for various purposes and to address different specialties in hydrology. It can be, for example, used for short-range forecasting, data assimilation, or multimodel studies. Because HOOPLA was not conceived for a specific application, it only includes basic functions for result visualization in order not to orient users toward a certain interpretation of the results. As a counterpart, the format of the input and outputs are heavily documented in the user manual.

Finally, to address the different sources of uncertainty located along the modelling hydrometeorological chain, HOOPLA adopts an approach based on ensembles. Because ensembles can be computationally expensive, a parallel computing feature that allows to run simultaneously several models and catchments has been implemented (see Appendix B).

## 2.2 Hydrological models

A collection of 20 lumped and conceptual hydrological models is provided in HOOPLA. These models perform rainfall-runoff transformation by the means of interconnected reservoirs (buckets) that fill and empty over time to simulate the hydrological processes. Additionally, the models assume that a catchment has homogeneous properties over its domain. Models were selected based on their:

- Complexity: only lowly to moderately complex models were selected providing that they perform well. Therefore, the retained models possess a rather low number of free parameters and reservoirs (see Table 1).

- Diversity: the models were developed by different research and engineering teams in various contexts and for different purposes. Differences in model structure is therefore fostered and should be reflected in dissimilar behaviors.





The selection originates from the work of Perrin (2000), where 35 models were tested in a context of model comparison to investigate the influence of the number of optimized parameters and model structure on the streamflow simulation performance. As some structures were deemed redundant, the number of models was subsequently reduced to 15 by Seiller et al. (2012) to which 5 more models were added in a multimodel mindset. Models have been coded according to Perrin (2000) appendix and original references are given in Table 1. As most models were coded based on theses, books, journals, or conference publications, this opens the door to some subjective programming.

**Table 1.** Main characteristics of the 20 lumped models

| Model name | Number of parameters | Number of reservoirs | Modif. | Derived from |
|---|---|---|---|---|
| HydroMod1 | 6 | 3 | Slightly | BUCKET (Thornthwaite and Mather, 1955) |
| HydroMod2 | 9 | 2 | Slightly | CEQUEAU (Girard et al., 1972) |
| HydroMod3 | 6 | 3 | Slightly | CREC (Cormary and Guilbot, 1973) |
| HydroMod4 | 6 | 3 | Slightly | GARDENIA (Thiery, 1982) |
| HydroMod5 | 4 | 2 | Similar | GR4H (Mathevet, 2005) |
| HydroMod6 | 9 | 3 | Substant. | HBV (Bergström and Forsman, 1973) |
| HydroMod7 | 6 | 5 | Slightly | HYMOD (Wagener et al., 2001) |
| HydroMod8 | 7 | 3 | Slightly | IHACRES (Jakeman et al., 1990) |
| HydroMod9 | 7 | 4 | Slightly | MARTINE (Mazenc et al., 1984) |
| HydroMod10 | 7 | 2 | Similar | MOHYSE (Fortin and Turcotte, 2007) |
| HydroMod11 | 6 | 4 | Similar | MORDOR (Garçon, 1999) |
| HydroMod12 | 10 | 7 | Substant. | NAM (Nielsen and Hansen, 1973) |
| HydroMod13 | 8 | 4 | Slightly | PDM (Moore and Clarke, 1981) |
| HydroMod14 | 9 | 5 | Substant. | SACRAMENTO (Burnash et al., 1973) |
| HydroMod15 | 8 | 3 | Slightly | SIMHYD (Chiew et al., 2002) |
| HydroMod16 | 8 | 3 | Substant. | SMAR (O'Connell et al., 1970) |
| HydroMod17 | 7 | 4 | Substant. | TANK (Sugawara, 1979) |
| HydroMod18 | 7 | 3 | Substant. | TOPMODEL (Beven et al., 1984) |
| HydroMod19 | 8 | 3 | Slightly | WAGENINGEN (Warmerdam et al., 1997) |
| HydroMod20 | 8 | 4 | Substant. | XINANJIANG (Zhao et al., 1980) |

We strongly emphasize that the hydrological models implemented in HOOPLA are not the original models nor their most up-to-date versions. In addition, the models were intentionally modified, mainly by Perrin (2000), to fit a common framework. For example, these changes may include a modification of the catchment spatial representation (all distributed models were converted to lumped models), a reduction of the number of free parameters, or discarding elements of the structure that did not address rainfall-runoff modelling (like potential evapotranspiration and snow accounting routine). Table 1 provides a qualitative





assessment of the changes that have been performed on the models, ranging from similar to substantially modified. A detailed list of modifications is provided in the HOOPLA user manual (Thiboult et al., 2018).

## 2.3  Snow accounting routine

The CemaNeige (Valéry et al., 2014) snow accounting routine (SAR) is implemented in HOOPLA to simulate snow accumulation and melt. It performs the partitioning of precipitation between liquid and solid phases by using a simple and parsimonious

approach based on 1) a spatial discretization of the catchment in altitudinal bands, 2) a transition temperature range that controls the liquid/solid fraction, 3) a term that describes the snowpack thermal inertia, and 4) a degree-day formula to estimate the snowmelt. CemaNeige results are subsequently used as inputs for the hydrological models, if necessary. CemaNeige is currently the only snow accounting routine included in HOOPLA as other tested routines did not bring satisfactory performance at this time.

## 2.4  Data assimilation schemes

Data assimilation aims at integrating information contained in observations to improve simulation accuracy while accounting for uncertainties in the measurements and model. More precisely, data assimilation schemes typically adjust the model states based on the latest available observations to provide better initial conditions for the next modelling step.

Two probabilistic, sequential data assimilation schemes are provided in HOOPLA:

– The Sequential Importance Resampling filter (SIR). It belongs to the class of particle filters (PF) and is also referred to as Sequential Monte Carlo or Bootstrap filter. More details about Bayesian inference and the PF derivation can be found in Arulampalam et al. (2002). The latter was the main inspiration for the SIR code implemented in HOOPLA.

– The Ensemble Kalman Filter (EnKF). A sound description of the EnKF is presented in Evensen (2003) and the EnKF code in HOOPLA was written according to Mandel (2006) recommendations.

These two sequential data assimilation schemes were selected to be included in HOOPLA because they have been extensively tested in hydrological sciences and proved to be efficient, which is reflected by their frequent use in the literature (Liu and Gupta, 2007). Even if both the PF and the EnKF adopt a Monte Carlo estimation method, they use a different approach and resort to different approximations.

## 2.5  Calibration algorithms

Calibration aims at identifying an optimal parameter set with regards to a chosen cost function (or efficiency criterion). HOOPLA provides two calibration algorithms: the Shuffle Complex Evolution (SCE, Duan et al., 1992) and the Dynamically Dimensioned Search algorithm (DDS, Tolson and Shoemaker, 2007). Both methods are iterative and global (they seek to find the optimal set among the entire parameter space). They mainly differ in the way the parameter space is sampled.

SCE is based on several groups of points (parameter sets called complexes) spanning the parameter space. At each iteration,

these complexes evolve along several mechanisms (namely reflection, contraction, and mutation) and are subsequently recom-





bined. These steps are iterated until the algorithm converges or a preset maximum of iterations is reached. The SCE is often considered as a state-of-the-art algorithm for high dimensional problems but remains computationally expensive.

DDS is inspired by the manual calibration procedure. The DDS explores the entire parameter space during the first optimization stages, and then progressively fixes parameter values, reducing the parameter space dimension. The parameter space is

sampled via random perturbations and reflections. A distinctive feature of the DDS is that it automatically scales the parameter search to find good solutions within a specified maximum number of iterations. As a result, it is a computationally efficient method that provides a better solution than the SCE when the number of model evaluations is limited. Therefore DDS is well suited for calibrating models with a large number of free parameters or that require important computing resources.

## 2.6 Potential evapotranspiration formulas

Potential evapotranspiration is used to quantify the atmospheric demand in water vapour and constitutes one of the hydrological model inputs (along precipitation). A large number of potential evapotranspiration formula exist. Only three were implemented in HOOPLA (see Table 2) because hydrological models usually express little sensitivity towards the PET input (Seiller and Anctil, 2016).

**Table 2.** Potential evapotranspiration formulas. Date refers to the calendar date, T, Tmin, and Tmax are respectively the mean, minimum, and maximum air temperature in $^{\circ}$C, Lat the latitude of the center of the catchment in $^{\circ}$, Rad the incoming solar radiation in $\mathrm{MJ/m^2/t}$ , Relhum the relative humidity in %, Wndspd the wind speed in $\mathrm{m/s}$, and z the average elevation of the catchment in m.

| Name | Required input data | Reference |
|---|---|---|
| Oudin | Date, T, Lat | Oudin et al. (2005) |
| Kharrufa | Date, T, Lat | Kharrufa (1985) |
| Penman | Date, T, Lat, Rad, Relhum, Tmax, Tmin, Wndspd, z | Penman (1948) |

Kharrufa and Oudin are parsimonious formulas in the sense that they both rely on an energy balance rationale that only
requires mean air temperature as input while the Penman formula is more complex and uses a combinational approach (energy balance and aerodynamic) that necessitates more data.

## 3 An application of HOOPLA

This section presents typical results obtained with HOOPLA in calibration, simulation, and forecast mode. Computations are performed with a 3-hour time step on 31 catchments situated in the southern part of the Province of Québec, whose rivers flow
into the Saint Lawrence river. The catchments area and the mean annual specific discharge range from 515 to 6839 $\mathrm{km^2}$ and from 1.1 to 3.4 $\mathrm{mm/day}$ respectively. The streamflow data were collected by the Direction de l'Expertise Hydrique du Québec





and are freely available on their website. Meteorological observations were made available by the Service de l'information sur le milieu atmosphérique (Bergeron, 2016) and ECMWF meteorological forecast were extracted from the TIGGE database (Bourgeault et al., 2010). Note that catchment 14 dataset is provided along the HOOPLA toolbox. Calibration, simulation, and

forecast are performed over the years 1997-2007, 2007-2015, and 2015-2017, respectively.

Because the watersheds undergo a nival influence, CemaNeige, the snow accounting routine included in HOOPLA, is activated. The Oudin formula (Oudin et al., 2005) is chosen among the available potential evapotranspiration formulas because of its low requirement of input data and known good performance when combined with the 20 aforementioned hydrological models at those sites. Finally, all 20 hydrological models provided with HOOPLA are used.

**Calibration**

In this example, the 20 hydrological models are calibrated individually with the Shuffle Complex Evolution algorithm (SCE, Duan et al., 1992) to maximize the modified Kling Gupta Efficiency score (Kling et al., 2012, KGE'). The SCE is parameterized such that the maximum number of iterations and the number of complexes are respectively set to 100 and 25. CemaNeige free parameters are not calibrated; the default HOOPLA three-hour time step values are used instead (snowmelt factor $\theta_{G1} = 0.4$

mm/°C/3h and cold-content factor $\theta_{G2} = 0.93$).

Figure 1 illustrates the performance values of the 20 models over the 31 catchments with the KGE'. Models perform generally well with an overall median KGE' close to 0.85. Performance varies more across catchments, which is expected considering that some are intrinsically harder to model. On this topic, the two catchments that exhibit a median performance below 0.6 are among the smallest ones (smaller than 800 $km^2$), which may indicate a possible limitation of HOOPLA regarding this

aspect.

**Simulation**

Simulation is carried out in combination with one of the data assimilation techniques provided in HOOPLA, namely the Ensemble Kalman Filter. HOOPLA parameters are defined in order to issue 50 members and to update the hydrological models every eight time steps, i.e. once a day. Following Thiboult et al. (2016), the EnKF hyperparameters are defined so

that the temperature uncertainty is assumed Gaussian with a standard deviation equal to 2 °C, the precipitation uncertainty is approximated by a Gamma distribution, which standard deviation is equal to 50 % of the observed precipitation, and the streamflow uncertainty is assumed Gaussian as well, with a standard deviation equal to 10 % of the observed streamflow. These hyperparameters are shared by every model and catchment.

Figure 2 confirms that HOOPLA provides a high level of performance over the catchments with an overall median KGE'

greater than 0.90. Data assimilation successfully reduces the simulation bias as shown by the global improvement of performance over the calibration. This suggests that an adequate data assimilation schemes may help compensate for a calibration that sought minimizing the error over an entire chronicle. This gain varies according to catchments, but also to models to a lesser extent (Thiboult and Anctil, 2015).





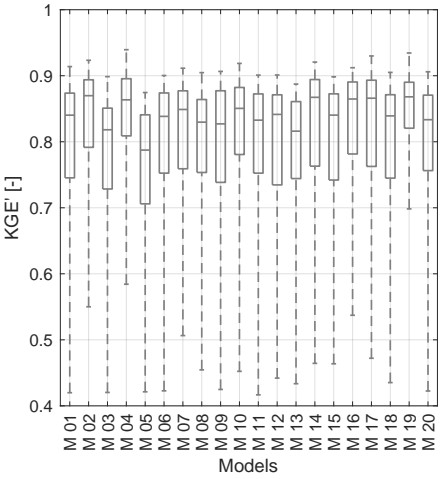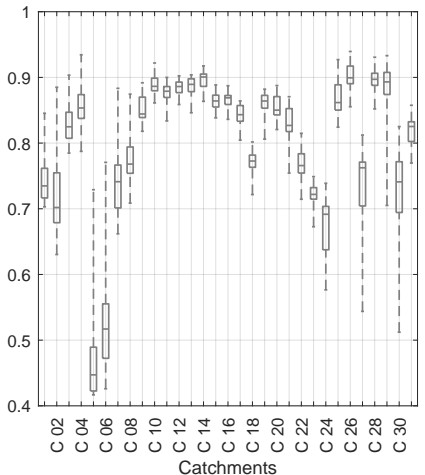

**Figure 1.** Performance in terms of modified Kling-Gupta Efficiency (KGE') of the 20 models in calibration over 31 catchments, by model (left) and by catchment (right)

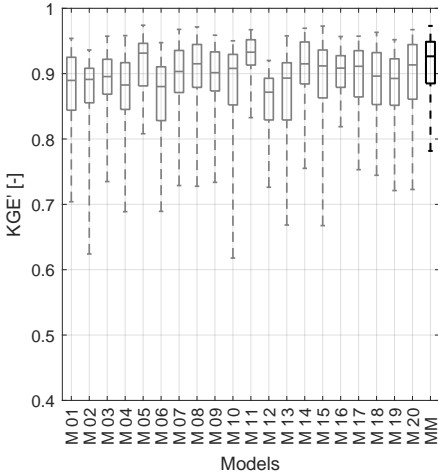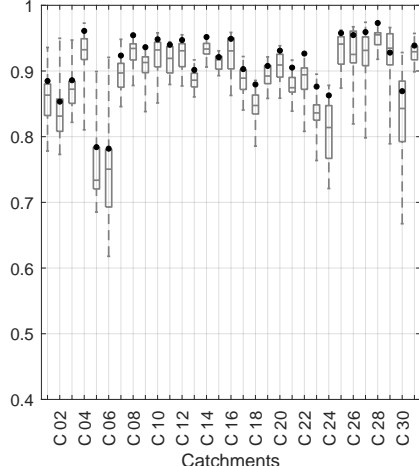

**Figure 2.** Performance of the 20 models and the multimodel (MM) in simulation over 31 catchments, by model (left) and by catchment (right). On the left plot, the multimodel is designated by the acronym MM, and by the black dots on the right plot. The Kling-Gupta Efficiency (KGE') is computed on the averaged EnKF members.

The multimodel, obtained by merging the individual simulations from the 20 hydrological models, performs generally well,
outperforming 18 of the 20 individual models.





**Forecast**

Forecast resorts to the same data assimilation setup than simulation. The 50-member meteorological ensemble forecast issued by the ECMWF is used as input for lead times up to 10 days. Therefore, each hydrological model ensemble is composed of 2500 members (as in Thiboult et al., 2016).

Figure 3 and 4 show that HOOPLA's ensemble prediction system is accurate for the 4th lead time (12 hours ahead) with most models' KGE' above 0.9. Similarly to the simulation, performance values in forecast vary more across catchments than models.

Forecast performance decreases with increasing lead time (Figure 3, 4, and 5). The loss of accuracy mainly comes from the decrease of the meteorological predictive skills and the fading influence of data assimilation. Yet, for the 80th lead time (10

days ahead), the median KGE' still lies around 0.55, indicating that the system remain an insightful predictor.

The multimodel approach thrives in short range forecasting and stands out from the other models as its median performance is greater for most lead times (Figure 5). It may be occasionally outperformed by individual models, but the latter differ from one catchment to another.

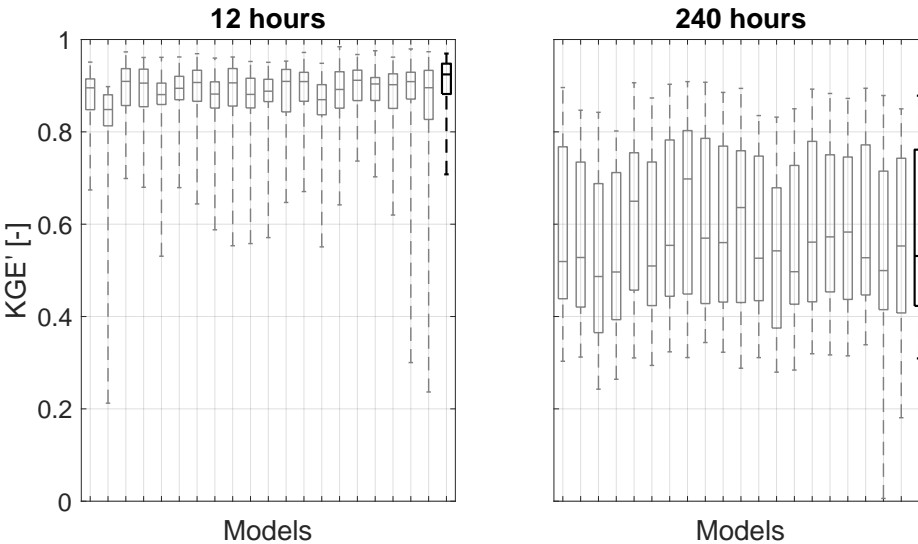

**Figure 3.** Performance of the 20 models and the multimodel (MM) in forecast with the EnKF, over the 31 catchments. The Kling-Gupta Efficiency (KGE') is computed on the averaged EnKF members.

## 4   Conclusions

This technical report presents the HydrOlOgical Prediction LAboratory (HOOPLA), a MATLAB toolbox that encompasses a broad range of functionalities to perform hydrological calibration, simulation, and forecast. The user may choose among a large





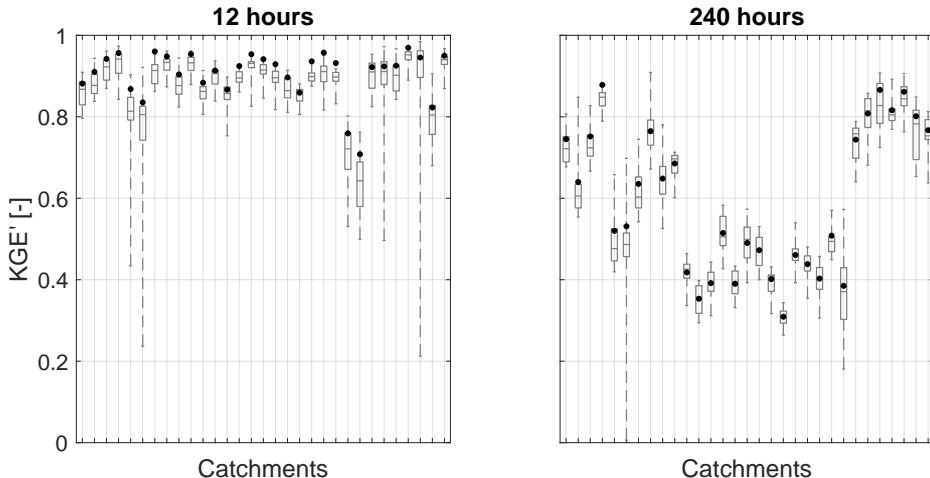

**Figure 4.** Performance of the 20 models in forecast with the EnKF, according to the 31 catchments. Multimodel results are depicted by black dots. The Kling-Gupta Efficiency (KGE') is computed on the averaged EnKF members.

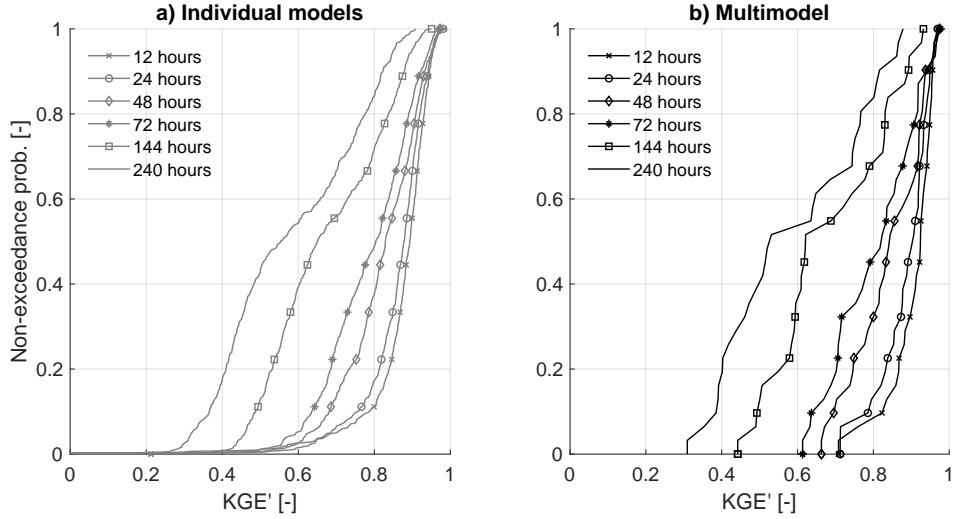

**Figure 5.** Performance in forecast according to the lead time. On the left, cumulative densities are constructed with the the Kling-Gupta Efficiency (KGE') of every model and catchment. On the right, cumulative densities are constructed with the KGE' of the multimodel on every catchment.



collection of tools and options to perform hydrological computations that fits one's needs. The toolbox includes among others 20 dissimilar lumped models, a snow accounting routine, three potential evapotranspiration formulas, two data assimilation schemes, two calibration algorithms, and parallel computing functionalities. Special attention was paid to create a framework

that can host additional tools, making HOOPLA a suitable platform to develop and test tools designed by the users. Future development will be driven by users' feedback and their contribution.

*Code and data availability.* The HOOPLA software, its manual, and a set of demo hydrometerological data are freely available under the BSD 2-Clause License at: https://github.com/AntoineThiboult/HOOPLA.

**Appendix A: The Launch_HOOPLA.m script**

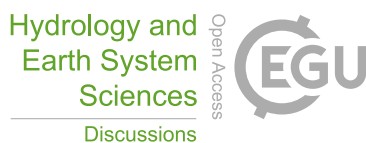

Launch_HOOPLA.m is HOOPLA's masterfile. It assigns values to a structure named "Switches" that translates user preferences into HOOPLA computations. For regular needs, this is the only file that needs editing. An extract of launch_HOOPLA.m is reproduced in Listing 1 to illustrate its structure. In addition to the comments written in this script, the meaning of the different switches is detailed in the user manual.

**Listing 1.** Organization of HOOPLA's masterfile (extract of launch_HOOPLA.m).

```
% This function is the main function which should be edited to specify
     % user's request. Multiple options are available for for the hydrological
     % calibration, simulation, and forecast. These options should be specified
     % in this file by modifying the values of the structure "Switches".
     ...
%% Section 1: Switches: 1=yes/on, 0=no/off [EDIT THIS SECTION]

     % Calibration / Simulation / Forecast
     Switches.calibration.on =1;  % Run calibration
     Switches.simulation.on  =1;  % Run simulation
Switches.forecast.on    =0;  % Run forecast

     % Dates:format 'yyyy/mm/dd/HH:MM:SS' for 3h time step, 'yyyy/mm/dd' for 24h time step
     Switches.calStart    = '1997/01/01/03:00:00' ;  % Beginning of calibration period
     Switches.calEnd      = '2007/01/01/00:00:00' ;  % End of calibration period
Switches.simStart    = '2007/01/01/03:00:00' ;  % Beginning of simulation period
     Switches.simEnd      = '2015/01/01/00:00:00' ;  % End of simulation period
     Switches.fcastStart  = '2015/01/01/03:00:00' ;  % Beginning of forecast period
     Switches.fcastEnd    = '2017/01/01/00:00:00' ;  % End of forecast period

% General switches
     Switches.timeStep           ='3h' ;  % Computation time step. Choose between '3h' and '24h'
     Switches.petCompute.on      =1;  % Compute PET
     Switches.snowmeltCompute.on =1;  % Compute snowmelt
     Switches.warmUpCompute.on   =1;  % Add warm up before modelling
Switches.verb.on            =1;  % Verbose. Display information about computing
     Switches.exportLight.on     =1;  % Export fewer data (/results) to save space
```



```
      Switches.overWrite.on           =1;       % Overwrite existing files created by HOOPLA
      Switches.parallelCompute.on =1;           % Parallel computing

      % Calibration switches
      Switches.calibration.export.on = 1;       % Export calibrated parameters to ./Data for future
         Simulation/Forecast once calibration is performed
      Switches.calibration.snowCal.on = 0;      % Calibrate snow module (if 0, default values are used)
      Switches.calibration.method ='SCE';       % Choose between 'DDS' and 'SCE'
Switches.calibration.rmWinter.on = 1;      % Remove the Quebec "ice months" (dec, jan, fev, mar)
      Switches.calibration.score ='NSE';        % Performance criteron (RMSE,MSE,NSE,etc. Enter "help det_score" in
         terminal to see all available scores)
Switches.calibration.maxiter = 1000;      % Maximum number of iteration during calibration
      Switches.calibration.SCE.ngs = 25;        % Number of Complexes for the SCE optimization

% Forecast switches
      Switches.forecast.issueTime        =6;    % Hour of the day for which a forecast is issued (can be several
         times per day ex: [6 12 18 24])
      Switches.forecast.perfectFcast.on  =0;    % Use meteorological observations as meteorological forecast
Switches.forecast.hor              =80;   % Horizon of the forecast (in time steps)
      Switches.forecast.metEns.on        =1;    % Use meteorological ensemble forecast
      ...
```





## Appendix B: Computation times

Table B1 provides an insight about computational time with a 3h time step. Tests were carried out over a 10-year period (29225 time steps) on a computer equipped with a processor Intel(R) Core(TM) i5-4210M clocked at 2.60 GHz and 8 Go of RAM. The experiment was repeated 100 times to minimize the influence of computer perturbations. As a reminder, HOOPLA offers a parallel computing option that may significantly decrease the time when multiple tasks are performed.

**Table B1.** Elapsed time for individual model runs, over a 10-year period (29225 time steps).

| Model | Min.(s) | Med.(s) | Max.(s) | Model | Min.(s) | Med.(s) | Max.(s) |
|-------|---------|---------|---------|-------|---------|---------|---------|
| M01 | 1.40 | 1.47 | 2.36 | M11 | 2.06 | 2.19 | 2.68 |
| M02 | 1.53 | 1.64 | 2.38 | M12 | 1.49 | 1.55 | 2.12 |
| M03 | 1.42 | 1.50 | 1.78 | M13 | 1.52 | 1.65 | 2.71 |
| M04 | 1.46 | 1.52 | 2.01 | M14 | 1.63 | 1.70 | 2.35 |
| M05 | 1.68 | 1.74 | 2.45 | M15 | 1.51 | 1.63 | 3.13 |
| M06 | 1.53 | 1.62 | 2.47 | M16 | 2.64 | 2.87 | 4.65 |
| M07 | 1.63 | 1.75 | 2.17 | M17 | 1.65 | 2.21 | 2.71 |
| M08 | 1.43 | 1.57 | 1.93 | M18 | 1.59 | 1.63 | 2.04 |
| M09 | 1.54 | 1.68 | 2.35 | M19 | 1.61 | 1.64 | 1.87 |
| M10 | 1.51 | 1.61 | 2.15 | M20 | 1.55 | 1.58 | 1.69 |

*Author contributions.* Conception and design of the work: A. Thiboult, G. Seiller, and F. Anctil. Data analysis and interpretation: A. Thiboult,
C. Poncelet, and F. Anctil. Drafting the article: A. Thiboult and C. Poncelet. Critical revision of the article: F. Anctil and G. Seiller. Final approval of the version to be published: A. Thiboult, G. Seiller, C. Poncelet, and F. Anctil.

*Competing interests.* The authors have no conflict of interest to declare.

*Acknowledgements.* The development of the toolbox was made possible by the financial support from the NSERC Canadian FloodNet initiative (Grant number: NETGP 451456). We would like to thank the Ministère du Développement durable, de l'Environnement et de
la Lutte contre les changements climatiques and more precisely the Direction de l'Expertise Hydrique du Québec and the Direction de l'Information sur le Milieu Atmosphérique for their collaboration and for allowing the diffusion of the demo hydrometeorological observation dataset provided with HOOPLA. We are also thankful to the TIGGE initiative for granting us permission to distribute a subset of their meteorological forecast database as demo forcing in HOOPLA. Finally, we express our great appreciation to the IRSTEA Antony (HYCAR team), and in particular Charles Perrin, for sharing over the years, their extensive knowledge about the hydrological models.





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
