# Peer review of "The HOOPLA toolbox: a HydrOlOgical Prediction LAboratory to explore ensemble rainfall-runoff modeling"

_Hydrology and Earth System Sciences, 2020_

## Referee Comment (RC1) · Shervan Gharari (Referee) · 8 Mar 2020

This technical note introduces a modeling framework developed in MATLAB environment for progressive model building, calibration and data assimilation. I welcome the contribution.

Most of my comments are regarding the manuscript and not the code itself. To me, it looks as if the manuscript is written in a rush! The abstract is not sufficiently explanatory of what is in the manuscript. It should be clear that the toolbox is evolving around simple bucket rainfall-runoff models. There is no consistency between the paragraphs of the first two pages describing the need for the toolbox. The model starts from

ensemble forecast, predictive uncertainty, initial condition, structural uncertainty and finally existing modeling toolboxes (where the manuscript refers to previous works on flexible toolboxes; also please include MARRMoT, Knoben et al., 2019). Given that, the manuscript does not position the toolbox well in the generation of other toolboxes that are designed to explore similar research question on structural comparison, input uncertainty, and data assimilations while it gives the feeling to readers that the proposed modeling framework is the only modeling framework that deals with input uncertainty while also considering the structural uncertainty combined. Following this comment, I would suggest:

1- Make the abstract more elaborative of what is going on in the manuscript. Possibly provide some understanding of the importance and the added values of HOOPLA.

2- Make sure that the paragraphs in the introduction are following logically. For example "In this respect" in the second paragraph should refer to the ensemble forecast from the first paragraph but instead, it explains the initial condition instead. Although they are related, the link here is not well explained and the story is not smooth.

3- Parameter and structural uncertainties are coming earlier than the framework which is designed to elaborate them (paragraphs 3 and 4 can be merged).

4- Section 2.1 and 2.2 can also be better organized. At this moment there is a mix of models structure, code capabilities, data assimilation, and optimization. I would suggest the authors separate the model and hydrological capabilities in one section, for example, 2.2 can be about the different hydrological models with the subsections of transpiration and snow, surface runoff formulation (section 2.3 and 2.6 can be put under the model section with more elaboration on the other processes as well if the authors wish to elaborate on the processes in this technical note), then calibration, and finally assimilation. Moreover, section 2.1 can be more technical rather than descriptive; the authors can objectively explain on parallelization, modularity in a much more structured fashion.

5- The information on the parallelization that is expressed in Appendix B should somehow come to section 2.1 as parallelization is one of the technical aspects of the toolbox. This section should really explain the toolbox capabilities clearly. There is model set up convenience provided by the toolbox but there should be also tangible gains provided by the toolbox as well. I would suggest Section 2.1 include subsection with the philosophy of development and also an actual feature of the toolbox such as parallelization, modularity, etc.

6- May be I missed here; what it the numerical implementation for the model. Forward explicit or iterative explicit or implicit? Also, explain more about the time stepping of the model. Can it do internal aggregation, interpolation of input forcing for example?

7- I think there are a lot of similar figures in the manuscript. I would suggest diversifying the figures by including an ensemble forecast for predictions for hydrograph for example. Including one figure with hydrograph would be a wise thing to do.

8- The conclusion should wrap up the technical aspect of the toolbox and its added values. It can be a bullet point conclusion and a short description of how those objectives have been met.

9- In my opinion, by looking at the code from GitHub, I feel that the codes can be written in a more efficient way using generic usable components. This is just a suggestion as I did not dive deep into the code. BTW, just mentioned the GitHub page once in the manuscript.

Overall, I would suggest a major revision for this manuscript. The presentation should be improved significantly.

With regards,

Shervan Gharari

---

## Referee Comment (RC2) · Anonymous Referee #2 · 16 Mar 2020

General comments

This technical note presents a toolbox for hydrological modelling and forecasting, based on a set of 20 hydrological models.

First, I found that the article looks like a user guide, not really like a scientific technical note. Many parts are very similar to the user-guide available online on the HOOPLA github space. The added-value of this paper compared to the toolbox user-guide seems limited.

Second, the authors do not explain the originality of the toolbox compared to other existing tools.

[Figure]

Third, I am unsure this is the right EGU journal to publish such a paper. As mentioned in the HESS guidelines for authors, "for manuscripts focused on the development and description of numerical models and model components, we recommend submission to the EGU interactive open-access journal Geoscientific Model Development (GMD)". So the authors may better submit their paper to GMD.

Last, I found that the article is too vague and imprecise. There is very limited justification of the choices that were made, with no guidance for the users. Many parts seem to have been written too concisely.

For these reasons (and others detailed below), I found that the article is not suitable for publication as a technical note in HESS.

Detailed comments

1. L5: "user-defined hydrometeorological tools". What do you mean here?

2. L17: Assimilation procedures can also be used for parameter updating. Is this possibility also offered by the toolbox?

3. L27: "FUSE"

4. L75: "in order not to orient users towards a certain interpretation of the results": What do you mean? A software should offer the proper tools to help the user to interpret results. If the focus is on ensemble prediction, the authors should explain how HOOPLA provides original tools to explore the quality and characteristics of the tested ensembles.

5. Section 2.2: The authors mention in the abstract that the models can be run at multiple time steps. How was this implemented? There are probably many model equations or parameters that are time-step dependent. How was this codded?

6. L88: It is unclear on which aspects the structural differences were evaluated.

7. L94: What do you mean by "subjective programming"?
8. Section 2.3: How many parameters are used in the snow routine presented here.

9. L105: How many altitudinal bands are used? Can this be parameterized?

10. L108-109: Is there any reference where these tests are explained? Why other snow routines do not work?

11. Section 2.4: Are the assimilation schemes applied to all models without any difference? I guess that, depending on the model used, not all the initial conditions of the various model stores are similarly important to update.

12. L127: Why were local methods not implemented in the toolbox? There are less demanding in terms of computing time.

13. L132: Can the optimization of 4 to 10 parameters be considered a "high-dimensional" issue?

14. L138: Do these models really require important computing resources? The computing times shown in the appendix are not that large.

15. L142: The sensitivity to potential evapotranspiration depends on the targeted variables. For low flows or long term water balance, the comment of limited sensitivity may not be valid. Besides, the authors should explain how these three formulas were selected. Is it because they perform well in different climatic environments?

16. L142: "PET" is not defined.

17. Tab. 2: Why are there two lines of inputs for the Kharrufa formula?

18. Section 3: A map of the catchment location could be introduced. A bit more could be said on the catchments (snow influences, human influences). Why a 3-hr time step was chosen? What is the range of response times for the test catchments?

19. L152: What is this website?

20. L155: Why dissimilar lengths of test periods were chosen?

21. L158: Is there any reference detailing previous results with this formula on these catchments?

22. L164: Why the parameters were not calibrated? How the default values of the parameters were determined?

23. L168-170: Is it a default of the toolbox or linked to the fact that precipitations are known with less accuracy on small catchments than on large ones? Several existing articles discuss this issue in the literature. Since a few models seem to obtain acceptable performance (>0.70), it may also be linked to the fact that there are some specific processes active on these catchments, which are well represented only by these few models.

24. L169-185: I did not understand what is called here "simulation". It seems that the models that were previously calibrated are now applied with data assimilation. But what for? The authors update model state using the observed flow to predict the flow at the same time step. Is it what is done and evaluated? What is the objective of this application? Why is not there any classical calibration-validation scheme?

25. L174: Why every 8 time steps? How was this chosen?

26. L175-177: How these levels of uncertainty were chosen?

27. L181-182: This sentence is not fully clear for me.

28. Fig. 1 (and others). Please indicate which quantiles are shown in the boxplots. Indicate in the caption the test period used.

29. L184: How the multi-model was built? Simply by considering a super-ensemble or was there any selection or combination technique used?

30. L187-188: Are the ECMWF at the 3-hour time step used or were there some pre-processing applied?

31. L195: "the system remain an insightful predictor": compared to what? The KGE' is

probably poorly informative in a forecasting context on the actual model performance. The assimilation technique artificially enhances the forecast efficiency, but the forecast may be of limited value compared to some poor-man forecast (typically persistence). Besides, performance in Fig. 3 (left) seems to be as high as in Fig. 2. Any comment on this?

32. Figs. 3-4: Please harmonize the layout with the previous figures to ease visual comparison.

33. Conclusion: Very short conclusion, lack of discussion. This should be improved.

34. L215: "available for the"

35. Appendix B: It would be useful to add the typical (e.g. median over the 31 catchments) number of model runs needed for calibration with the SCE and DDS algorithms.

---

## Author Comment (AC1) · 17 Apr 2020

We would like to thank Shervan Gharari and an anonymous reviewer for reading our manuscript and for their comments. While we would agree to modify our manuscript according to the reviewers' recommendations, we also realized, thanks to the referee #2, that HESS discourages manuscripts focused on the development and description of numerical models. Thus, we will resubmit this paper in Geoscientific Model Development.